# Genome-Wide Association Study of Breast Density among Women of African Ancestry

**DOI:** 10.3390/cancers15102776

**Published:** 2023-05-16

**Authors:** Shefali Setia Verma, Lindsay Guare, Sarah Ehsan, Aimilia Gastounioti, Gabrielle Scales, Marylyn D. Ritchie, Despina Kontos, Anne Marie McCarthy

**Affiliations:** 1Perelman School of Medicine, University of Pennsylvania, Philadelphia, PA 19104, USAmarylyn@upenn.edu (M.D.R.);; 2Washington University School of Medicine in St. Louis, St. Louis, MO 63130, USA; 3Spelman College, Atlanta, GA 30314, USA

**Keywords:** genome-wide association study, breast density, breast cancer, African ancestry

## Abstract

**Simple Summary:**

In the US, Black women are disproportionately affected by higher breast cancer mortality rates and later-stage tumor diagnoses compared with White women. Breast density, the ratio of dense fibroglandular breast tissue to overall breast tissue area, has previously been identified as an important breast cancer risk factor. Most current genome-wide association studies for breast density have been performed in participants of European ancestry, which have yielded important insights into genetic etiology of breast density. However, little is known about the influence of common genetic variants on breast density in African ancestry populations. Our study aimed to determine genetic factors associated with breast density in African ancestry populations using a Genome-Wide Association Study (GWAS) and post-GWAS analyses on a cohort of genomic data available through the Penn Medicine BioBank. Results of this study elucidate potential genetic mechanisms associated with breast density, and thus cancer risk, among women of African ancestry.

**Abstract:**

Breast density, the amount of fibroglandular versus fatty tissue in the breast, is a strong breast cancer risk factor. Understanding genetic factors associated with breast density may help in clarifying mechanisms by which breast density increases cancer risk. To date, 50 genetic loci have been associated with breast density, however, these studies were performed among predominantly European ancestry populations. We utilized a cohort of women aged 40–85 years who underwent screening mammography and had genetic information available from the Penn Medicine BioBank to conduct a Genome-Wide Association Study (GWAS) of breast density among 1323 women of African ancestry. For each mammogram, the publicly available “LIBRA” software was used to quantify dense area and area percent density. We identified 34 significant loci associated with dense area and area percent density, with the strongest signals in *GACAT3*, *CTNNA3*, *HSD17B6*, *UGDH, TAAR8, ARHGAP10*, *BOD1L2,* and *NR3C2*. There was significant overlap between previously identified breast cancer SNPs and SNPs identified as associated with breast density. Our results highlight the importance of breast density GWAS among diverse populations, including African ancestry populations. They may provide novel insights into genetic factors associated with breast density and help in elucidating mechanisms by which density increases breast cancer risk.

## 1. Introduction

Black women in the US have 40% higher breast cancer mortality than White women [1] and are more likely to be diagnosed with later stage tumors and with triple negative breast cancers, which have limited treatment options and poorer prognosis than hormone receptor positive tumors [2,3]. Given these disease patterns, early detection is vitally important for reducing racial disparities. Breast density, the relative amount of fibroglandular versus fatty breast tissue, is one of the strongest breast cancer risk factors, and increases breast cancer risk 3–5-fold [4,5]. Therefore, breast density is an important factor to consider in order to identify women at high risk who may benefit from intensified or supplemental screening. Black women have lower breast density on average than White women when assessed visually by the radiologist [6,7]. This is partly due to the fact that Black women tend to have higher body mass index (BMI) than White women [8], resulting in a greater relative amount of fatty breast tissue leading to radiologists’ lower assessments of density levels. Advances in breast imaging and quantitative image analysis are enabling the development of novel breast density metrics that are more precise and reproducible than the radiologist visual assessment of density [9,10,11]. Our group has developed novel computational methods to measure breast density from breast images [9,10], and our previously published work found that Black women had higher breast density on digital mammography (DM) than White women when breast density was measured quantitatively and adjusted for BMI [6].

The mechanisms that underlie the association of breast density with higher breast cancer risk are poorly understood. Breast density is highly heritable, with family-based studies estimating that 50–70% of the variance in breast density is explained by genetics [12,13]. To date, 50 genetic loci have been associated with breast density, however, these studies have been performed among predominantly European ancestry populations [12,14,15]. Understanding genetic factors associated with breast density, particularly among Black women, may help in identifying mechanisms by which breast density increases breast cancer risk as well as mechanisms that contribute to racial disparities in breast cancer. In the current study, we performed a Genome-Wide Association Study (GWAS) to identify SNPs associated with quantitative area-based measures of breast density among Black women.

## 2. Materials and Methods

### 2.1. Study Data

We utilized a cohort of women aged 40–85 years who underwent mammography screening (Selenia or Selenia Dimensions; Hologic Inc, Marlborough, MA, USA) at the Hospital of the University of Pennsylvania from 1 September 2010 through 31 December 2014 and did not have a known *BRCA1/2* mutation (N = 32,213 screening exams). We excluded screening exams with uncertain outcomes (N = 13), screenings preceded by a breast cancer diagnosis (N = 74), and true positive and false negative screening exams (N = 153). We additionally excluded screenings where a breast implant was present (N = 429), screenings for which all four processed (i.e., ‘FOR PRESENTATION’) DM views were unavailable (N = 406), and screenings for which breast density evaluation software failed (N = 808). Then, we removed screenings for women who did not consent to participate in the Penn Medicine BioBank (PMBB), a research biorepository where patients provide a blood sample and broadly consent to allow their biospecimens to be used for research purposes (N = 22,627). We excluded screenings for women without a genetically-informed genotype (N = 3718), and women who did not self-identify as Black or have an African ancestry genotype (N = 1803). From this pool of screening exams, we selected the earliest exam per person, resulting in 1348 individuals.

Women completed a breast cancer risk factor questionnaire at the time of mammography, from which reproductive factors, age, height, and weight were pulled. Body mass index (BMI) was calculated from self-reported height and weight, supplemented with electronic health record (EHR) data by using the nearest available measurement within 1 year prior to and 6 months after the date of screening. Implausible BMI values (<12 kg/m^2^ or >82 kg/m^2^) [16] were set to missing. Women were considered as postmenopausal if their menstrual periods had stopped or if they were over the age of 55. Patients still missing data on BMI, menopausal status, or age after this were excluded from the analysis (N = 15), resulting in a final study population of 1333 women.

### 2.2. Breast Density

Breast Imaging Reporting and Data System (BI-RADS) 4th or 5th edition breast density was obtained from the mammography report. For each mammogram, the publicly available “LIBRA” software (v1.0.4) was used to automatically quantify breast density [10,17]. Briefly, LIBRA partitions the breast region into density clusters of similar gray-level intensity, which are then aggregated into the final dense tissue segmentation. Summing the area of dense pixels provides total absolute dense area (DA), and normalizing DA by the total breast area gives area percent density. We used the dense area and area percent density estimates obtained from all mammography views available for each woman. A per-woman value of each density measure was generated by averaging the corresponding density estimates from all breast views. Since each view is only a 2D cross section of the breast, none of the views independently capture the true volume of density in the breast. By averaging estimates across each view and across breasts for each woman, we produce a more robust estimate of the actual density in the breast. Distributions of breast area, dense area, and area percent density were visually inspected, and observations that fell greater than 3 standard deviations above the mean were excluded (3 for breast area, 19 for dense area, and 27 for area percent density out of 1333).

### 2.3. Genome-Wide Association Study in Penn Medicine BioBank

We performed a GWAS on our cohort of 1333 women using PLINK 2.0 [18], a tool that employs a generalized linear regression model approach to association testing. We applied the following filters to the imputed PMBB data of the individuals: Excluded 189 individuals to remove relatedness, kept variants with Hardy–Weinberg equilibrium value greater than or equal to 10^−6^, and only kept variants with minor allele frequency greater than 0.01 [19,20]. The covariates used in the model were BMI, age, two principal components of genetic ancestry [21,22], and menopausal status (binary). Principal component analysis (PCA) was conducted using fast PCA approximation in EIGENSOFT package by projecting PCS on 1000 Genome population [23]. Appendix A shows PCA plot for PC1 and PC2 and scree plot showing proportion of variance explained by first 10 PCS to identify significant PCs to use as covariates in this study. For all GWAS results with a *p*-value of 10^−5^ or less, we used Biofilter [24,25] to annotate the variants with their nearest genes. To identify significant and suggestive loci, we applied the clumping parameter in PLINK, which involved identifying the lead single nucleotide polymorphism (SNP) within a 100 kilobase (KB) window of SNPs with a linkage disequilibrium (LD) threshold of R2 ≥ 0.2, as the SNP that represented each locus.

### 2.4. Functional Mapping

We used polyfun [26] for functionally informed fine-mapping. To define functional variants, we used ENSEMBL sequence annotations and epigenetic annotations from EpiMap breast tissue samples [27]. We made the annotations disjoint to optimize regression stability, prioritizing smaller categories. Hapmap SNPs were selected for the regression weights for the L2-regularized h2 step. For the fine-mapping step, we set a *p*-value threshold of 10^−4^ and a maximum number of causal variants of 10. Then, we searched for any variants with a posterior probability of greater than or equal to 0.5 for further annotation.

### 2.5. Transcriptome-Wide Association Study (TWAS)

We performed summary-based TWAS for area percent density and breast dense area [28]. The S-PrediXcan best practices workflow was used to impute expression levels based on GTEx [29] models and to test association in breast mammary tissue only.

### 2.6. GWAS-Catalog Lookup

We performed a lookup of all nominally significant SNPs and genome-wide significant genes identified from GWAS as described in Section 2.3 in the EMBL-EBI GWAS catalog (summary statistics downloaded on 1 October 2022) [30] to identify the known associations of SNPs and genes from our study with other phenotypes. We filtered the catalog results to associations with a reported *p*-value of no more than 5 × 10^−8^.

### 2.7. Correlation Analyses

We performed estimation of SNP-based heritability and genetic correlation between breast density traits, age, BMI, and menopause status using GCTA’s Haseman–Elston’s regression approach [31,32]. Additionally, we compared effect sizes and MAF for significant variants from recently published breast density GWAS in European ancestry population [14] with estimates from GWAS in our study of African ancestry participants only.

## 3. Results

The characteristics of the study population (N = 1333) are displayed in Table 1 and compared with the total underlying cohort. The average age at screening was 56.7, most patients were postmenopausal (72.2%), and just over 12% had a family history of breast cancer. Based on radiologist-rated BI-RADS density, 18% had heterogeneously dense breasts and 0.5% had extremely dense breasts. The mean area percent density was 22.5% (SD 11.1%).

### 3.1. Genome-Wide Association Analyses

In our quantitative GWAS consisting of women of African ancestry and/or identifying as Black, we discovered sixty-five significant hits consisting of twenty-nine independent loci (*p* < 5 × 10^−8^) for dense area and nine significant SNPs consisting of five independent loci for area percent density, as shown in Table 2 and Figure 1. No genome-wide significant associations for breast area were identified.

The strongest signals were in genes GACAT3 (*p* = 6.06 × 10^−13^), CTNNA3 (*p* = 1.76 × 10^−12^), HSD17B6 (*p* = 4.02 × 10^−12^), UGDH (*p*-value = 1.45 × 10^−11^), TAAR8 (*p* = 1.08 × 10^−10^), ARHGAP10 (*p*-value = 1.40 × 10^−10^), BOD1L2 (*p* = 4.17 × 10^−10^), and NR3C2 (*p* = 4.68 × 10^−10^). Two of these genes, CTNNA3 [33] and PRIM1 [34], have been previously associated with breast cancer. Variants in genes LRP1B (*p* = 1.56 × 10^−8^), TLL1/SPOCK3 (*p* = 4.41 × 10^−9^), DELEC1 (*p* = 3.84 × 10^−8^), LINC00858 (*p* = 1.01 × 10^−8^), CCSER2 (*p* = 1.59 × 10^−8^), and ZIC5 (*p*-value = 2.71 × 10^−8^) were associated with area percent density. We mapped all SNPs with *p*-value < 1 × 10^−5^ to the closest genes (Figure 1). This identified thirty-one unique genes, of which five have previously been associated with breast cancer (LGR6, NR3C2, LOC105377563, CTNNA3, PRIM1). None of these genes have been previously associated with breast density.

Complete summary statistics for all loci at suggestive *p*-value threshold of 1 × 10^−5^ are reported in Appendix A. Our results identified 60 unique loci that have been previously associated with breast cancer and replicated 13 genes that have been associated with breast density in other studies [12,14,15]. Quantile–quantile plots for area percent density, dense area, and breast area GWAS are displayed in Figure 2. The genomic inflation factor for area percent density was 1.01, dense area was 1.002, and breast area was 1.004, indicating little evidence of systematic error, such as population stratification.

### 3.2. Functional Mapping

Seventeen variants for breast density and eighteen variants for dense area are identified through fine-mapping at posterior probability >0.98. Among the fine-mapped results are the variants mapped to Enhancer regions on genes, such as KIFC3, CNGB1, and heterochromatin regions on genes PDE10A and KIFC3, as shown in Table 3.

### 3.3. Transcriptome-Wide Association Analyses

For each breast tissue TWAS, 14k genes were tested; therefore, the corrected *p*-value threshold is 3.6 × 10^−6^. Genes with a *p*-value of less than or equal to 1 × 10^−3^ are highlighted in the TWAS plots (Figure 3). Appendix A contains summary statistics from TWAS analyses. For dense area, one variant on chromosome 12 (rs1877183750) maps to *PRIM1* gene in dense area GWAS and is associated with expression of *CD63* (*p* = 1.9 × 10^−6^, Z = −4.75, Figure 3a). For area percent density, one variant on chromosome 19 (chr19:35115879) is associated with expression of *FXYD3* (*p* = 7.8 × 10^−7^, Z = −4.93, Figure 3b). *CD63* [35,36,37,38,39], *PRIM1* [34], and *FXYD3* [40,41] have been previously implicated in breast cancer.

### 3.4. GWAS Catalog Lookups

In the EMBL-EBI GWAS catalog, we found many traits that had reported associations with the genes (Appendix A) that were found to be significantly associated with area percent density and dense area. Traits associated with three genes were cardiovascular diseases, lipid measurements, immune system diseases, body mass index, and response to drugs. All traits shown in Figure 4 were associated with at least two genes.

Our set of suggestive SNPs from all three GWAS (*p* < 1 × 10^−5^) was reduced to only eight SNPs when we looked for an overlap with reported associations in the GWAS catalog, as shown in Figure 5. One SNP was associated with breast size (*rs10110651*), rs61895110 was associated with bone density, and *rs77754964* was associated with FEV/FEC ratio (measurement of forced expiratory volume). Other SNP-trait pairings are: *rs78049001*, cognitive decline; *rs2976530,* hip bone mineral density; *rs75986475*, physical activity; *rs78730126*, sex hormone binding globulin measurement.

### 3.5. Correlation Analyses

Genetic correlation among the breast density traits evaluated in this study are shown in Figure 6. The results suggest positive correlation between all breast density traits. Positive correlation was also observed between BMI and the three breast density traits. However, a negative correlation is observed between BMI and menopause status. Genetic correlation between menopause status and breast density traits was close to 0.

### 3.6. Comparison among EUR and AFR Breast Density GWAS

We compared our results with effect estimates and significance reported in a recently published breast density GWAS for 27,900 European ancestry individuals [14] (Table 4). Two SNPs identified as genome-wide significant in EUR ancestry study were significant at *p*-value < 0.001 in our analyses (rs16885613 and rs10087804).

## 4. Discussion

To our knowledge, ours is the first GWAS of quantitative breast density measurements performed among women of African ancestry. Among 1333 women, we measured dense area and area percent density from digital mammograms using a validated software algorithm and found sixty-five variants in twenty-nine genes associated with dense area and nine variants in five genes associated with area percent density. Our results highlight the potential value of examining SNPs associated with breast density among women of African ancestry, emphasizing the need for diverse ancestry analyses to better understand the genetic underpinnings of breast density and its impact on breast cancer risk in underrepresented populations.

Of the loci identified in this study, 13 of these regions had been previously identified as associated with breast density in studies of European ancestry populations [12,14,15]. Fifty-seven loci had previously been identified as associated with breast cancer risk among European ancestry populations [42], and three had previously been associated with breast cancer risk among African ancestry populations [43].

Several of the identified SNPs and genes have potentially plausible mechanistic connections to breast density and breast cancer risk. Two SNPs were identified with genome-wide significance in *CTNNA3*, catenin alpha 3, which encodes a protein involved in cell–cell adhesion in muscle cells. *CTNNA3* was previously identified in breast cancer GWAS [33]. Alpha and beta catenins have been implicated in cancer cell metastasis [44]. Additionally, prior African ancestry GWAS found *CTNNA3* to be associated with metabolic syndrome. This is interesting given the observed differences in breast density by BMI levels [6]. In addition, we identified a variant within a gene encoding another alpha catenin, *CTNNA1*, which has recently been categorized as a predisposition gene for Hereditary Diffuse Gastric Cancer [45]. Furthermore, loss of function mutations have been identified among breast cancer patients undergoing multigene panel testing [46]. Together, these findings suggest further research on the role of alpha catenins in both breast density as well as breast cancer risk.

*HSD17B6,* hydroxysteroid 17-beta dehydrogenase 6, is involved in androgen catabolism and has been implicated in polycystic ovarian syndrome (PCOS) [47], including metabolic perturbations correlated with PCOS, including increased BMI, fasting insulin, and insulin resistance [48]. LRP1B encodes a member of the low density lipoprotein (LDL) receptor family, which has been implicated in both metabolic phenotypes [49] and several cancers [50]. *GACAT3*, gastric cancer associated transcript 3 is a long non-coding RNA that has been previously implicated in gastric and other cancers, with high expression observed in breast cancer tissue [51] and correlated with prognosis among breast cancer patients [52].

Fine-mapping results include the heterochromatin region of the phosphodiesterase 10A (*PDE10A*). PDEs have oncogenic effects, and several preclinical studies have shown that inhibition of PDEs has an anti-tumor effect [53]. A recent study demonstrated that inhibition of *PDE10A* decreased cell proliferation, induced cell cycle arrest, and increased apoptosis in ovarian cancer cells [53]. Kinesin family member C3 (*KIFC3*) encodes a member of the kinesin-14 family of microtubule motors. These motor proteins attach to microtubules and move along them to transport cellular cargo. Overexpression of *KIFC3* was shown to be associated with resistance to docetaxel in breast cancer cell lines [54]. *SH3GL3* has been implicated as a tumor suppressor in glioblastoma and lung cancer, as well as in cell migration and invasion in myeloma, and has been previously identified in colorectal cancer GWAS associated with colorectal cancer [55].

Exploratory TWAS identified two loci associated with dense area and percent density, *CD63*/*PRIM1* and *FXYD3*. CD63 encodes a membrane protein of lysosomes [38] and glycosylation of this protein has been shown to affect breast carcinogenesis [35,36,37,38]. In addition, *CD63* was identified when machine learning was applied to GWAS data with respect to radiation-associated contralateral breast cancer [39]. *PRIM1*, which encodes DNA primase polypeptide 1, has been found to be overexpressed in breast tumors [34]. The observed association with *PRIM1* may also be explained by the fact that *PRIM1* is associated with age at menopause, and breast density is known to decrease following menopause [56]. *FXYD3*, an mRNA also known as *Mat-8* (Mammary tumor 8 kDa), is highly expressed in breast cancers [40] and has been shown to regulate breast cancer stem cells [41].

Despite the modest sample size, our study identified novel SNPs with plausible mechanistic connections to both breast density and breast cancer risk. Breast density was measured quantitatively using an automated algorithm with high accuracy. In addition, breast density is known to be highly heritable. In combination, the continuous quantitative trait and the high heritability may have resulted in the ability to detect moderate-to-large associations despite a relatively small sample size. However, the small sample size limits our ability to detect more modest associations. Therefore, replication in larger populations of African ancestry populations and meta-analyses are warranted to increase statistical power to detect additional SNPs with more modest associations with breast density.

Given that the biological underpinnings of breast density are poorly understood, further investigation of our findings may help in identifying pathways relevant for breast density development as well as the mechanistic relationship between breast density and breast cancer risk. In this study, we investigated the overlap between our top-associated variants and those reported in previous studies of breast cancer and other related traits. We believe that this integrative approach can help in shedding light on the underlying biology of breast density and its relationship to breast cancer risk. Future research is needed to validate our preliminary findings and further explore the functional implications of the identified genetic variants.

Our study highlights the value of exploring the genetic factors associated with breast density among African ancestry populations, providing proof-of-concept that additional SNPs relevant to breast density may be identified through expanding the diversity of GWAS studies. Furthermore, a strength of our study is its use of quantitative measures of breast density, which have been shown to be strongly correlated to breast cancer risk [9,10] and more objective and reproducible than radiologist-rated breast density [11]. Despite these strengths, our sample size was small, and therefore results will need validation in larger studies. Furthermore, novel fully volumetric methods derived from digital breast tomosynthesis, or 3D mammograms, may provide even more precise quantification of dense breast tissue enabling even greater power to detect associations with genetic factors.

## 5. Conclusions

We report the first GWAS of breast density among women of African ancestry, in which we identified novel SNPs associated with quantitative breast density measures, many of which had been previously identified as associated with breast cancer. Our results mark the beginning of the study of breast density among African ancestry populations and provide hypothesis generating findings that may help in clarifying the biology of both breast density and breast cancer.

## Figures and Tables

**Figure 1 cancers-15-02776-f001:**
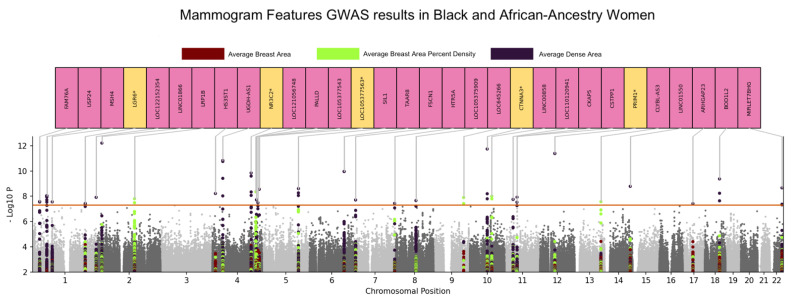
A composite Manhattan plot for area percent density, dense area, and breast area GWAS. X-axis shows chromosome position and y-axis represents the −log10 (*p*-values), the colors of all annotated loci are demonstrating the breast density phenotype tested. The SNPs mapping to closest genes are annotated in the plot where *p*-value < 1 × 10^-5^. Yellow background color and asterisk next to the gene name refers to previously known associations with breast density or breast cancer and pink background color refers to novel associations. For each gene, lowest *p*-value on SNP mapping to the respective gene is reported in the annotations in pink and yellow.

**Figure 2 cancers-15-02776-f002:**
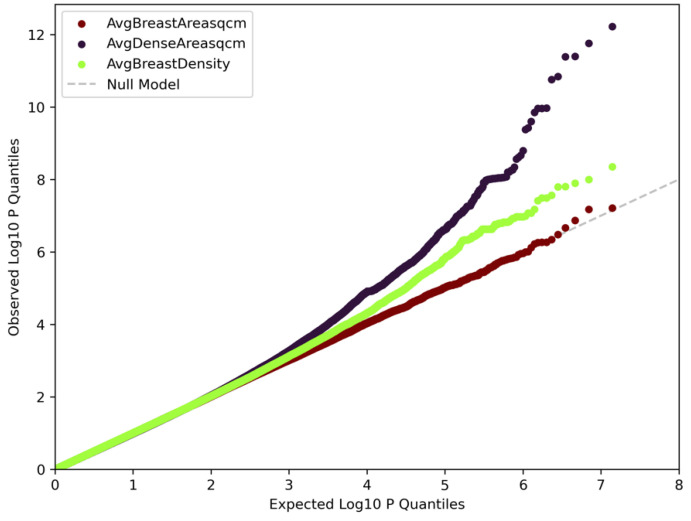
Quantile–quantile plots for area percent density, dense area, and breast area GWAS. Composite QQ plots for area percent density (green), dense area (dark purple), and breast area (dark red) GWAS. Genomic inflation factor for area percent density is 1.01, dense area is 1.002, and breast area is 1.004.

**Figure 3 cancers-15-02776-f003:**
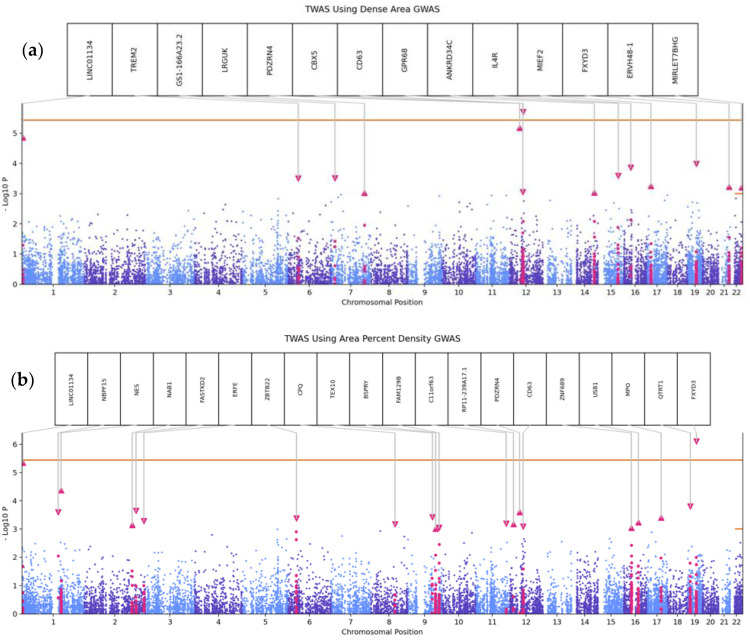
Transcriptome-wide association study analyses results for dense area (**a**) and area percent density (**b**)**.** Scatterplots representing chromosome positions of all tested SNPs on x-axis and −log10 (*p*-value) from TWAS analyses using breast tissues on y-axis. Genes at *p*-value < 1 × 10^−3^ are annotated. The point of the pink triangles represents direction of effect.

**Figure 4 cancers-15-02776-f004:**
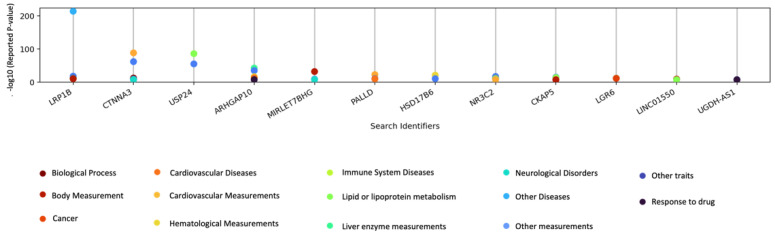
PheWAS of significant genes from mammogram trait GWAS. Scatter plot representing gene names on x-axis and −log10 (*p*-value) from GWAS catalog on y-axis. Each colored point corresponds to a different disease/disease category from GWAS catalog.

**Figure 5 cancers-15-02776-f005:**
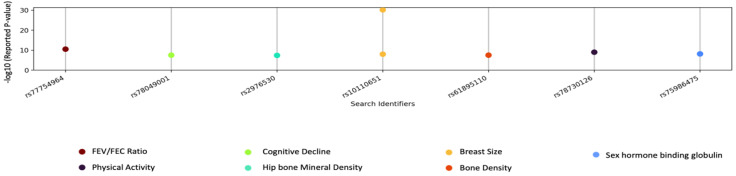
Suggestive SNPs from mammogram trait GWAS. GWAS catalog lookup results for suggestive SNPs on x-axis and −log10 (*p*-value) from GWAS catalog on y-axis. Each colored point corresponds to a different disease/trait.

**Figure 6 cancers-15-02776-f006:**
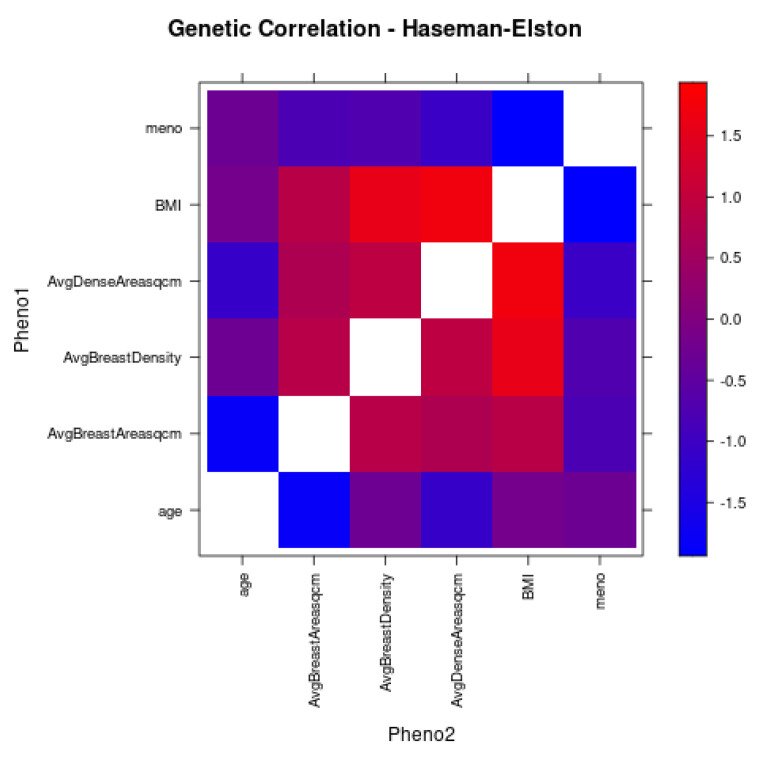
Heatmap showing genetic correlation between breast density traits, age, BMI, and menopause status (meno). The blue color corresponds to negative correlation and red corresponds to positive correlation.

**Table 1 cancers-15-02776-t001:** Characteristics of Mammography Screening Population. The characteristics of the screening population were compared with the total cohort. The average screening age was 56.7, most patients were postmenopausal, and about 12% had a family history of breast cancer.

Characteristic	All Self-Reported Black or African American Women, N = 10,090 ^1^	Study Cohort, N = 1333
Breast Area (cm^2^), Mean (SD)	206.71 (81.80)	217.63 (85.08)
Area Percent Density (%), Mean (SD)	13.89 (11.40)	11.52 (6.65)
Dense Area (cm^2^), Mean (SD)	26.30 (28.28)	22.52 (11.06)
Age, Mean (SD)	56.45 (10.57)	56.65 (9.80)
Atypical Hyperplasia, n (%)	26 (0.26)	4 (0.3)
Body Mass Index (Kg/m^2^), Mean (SD)	32.44 (7.40)	33.87 (7.85)
Menopausal Status, n (%)		
Postmenopausal	6926 (68.64)	963 (72.2)
Premenopausal	3164 (31.36)	370 (27.8)
Breast Density BI-RADS Categories, n (%)		
Almost entirely fat	1810 (17.94)	269 (20.2)
Scattered fibroglandular tissue	6053 (59.99)	814 (61.2)
Heterogeneously dense	2111 (20.92)	240 (18.0)
Extremely dense	91 (0.90)	7 (0.5)
Missing/Unknown	25 (0.25)	3 (0.2)
Cancer Diagnosis After Screening, n (%)	171 (1.69)	33 (2.5)
Number of Biopsies, n (%)		
None	8206 (81.33)	1053 (79.0)
One	1489 (14.76)	217 (16.3)
Two or more	395 (3.91)	63 (4.7)
Age at Menarche, n (%)		
<12 years	2129 (21.10)	318 (23.9)
12–13 years	4358 (43.19)	550 (41.3)
14+ years	2239 (22.19)	306 (23.0)
Unknown	1364 (13.52)	159 (11.9)
Age at First Live Birth, n (%)		
No Births	1495 (14.82)	187 (14.0)
<20	3536 (35.04)	524 (39.3)
20–24	2504 (24.82)	354 (26.6)
25–29	1272 (12.61)	145 (10.9)
30+	650 (6.44)	72 (5.4)
Birth age missing	633 (6.27)	51 (3.8)
Family History of Breast Cancer, n (%)		
None	8788 (87.10)	1171 (87.8)
One relative	1147 (11.37)	145 (10.9)
Two or more relatives	155 (1.54)	17 (1.3)

^1^ Mean (SD); n (%).

**Table 2 cancers-15-02776-t002:** Significant Loci in Genome-Wide Association Analyses. In sixty-five hits consisting of twenty-nine independent loci from genome-wide significant results of area percent density and five loci associated with dense area were discovered. #SNPs in loci refer to count of SNPs, which are in column SNP2.

Snpid	Variant_ID	Trait	Additional #SNPS in Loci	SNP2	Gene	*p*-Value
RS75413938	chr1:55151613:T:G	Dense Area	29	1:55066599(1),1:55069787(1),1:55078520(1),1:55079038(1),1:55079364(1),1:55091687(1),1:55096857(1),1:55099538(1),1:55103811(1),1:55108626(1),1:55120773(1),1:55126542(1),1:55128390(1),1:55128925(1),1:55128926(1),1:55134044(1),1:55155440(1),1:55161852(1),1:55167712(1),1:55177218(1),1:55183862(1),1:55189932(1),1:55193134(1),1:55194898(1),1:55195273(1),1:55195399(1),1:55203473(1),1:55217682(1),1:55226938(1)	*USP24*	8.92 × 10^−9^
RS79314581	chr1:244097544:G:C	Dense Area	0	NONE	*LOC122152354*/ *RN7SL148P*	1.22 × 10^−8^
RS190395094	chr1:27735235:C:T	Dense Area	8	1:27646562(1),1:27649415(1),1:27657130(1),1:27674922(1),1:27694043(1),1:27725469(1),1:27820501(1),1:27823924(1)	*FAM76A*	2.64 × 10^−8^
RS60005977	chr1:75838225:C:T	Dense Area	0	NONE	*MSH4*	2.77 × 10^−8^
RS78168242	chr1:202232716:C:T	Dense Area	2	1:202217643(1),1:202220169(1)	*LGR6*	3.73 × 10^−8^
RS142447005	chr2:16941818:G:T	Dense Area	0	NONE	*LINC01866*	6.06 × 10^−13^
RS139721819	chr4:39529250:C:G	Dense Area	6	4:39487863(1),4:39494788(1),4:39497063(1),4:39523173(1),4:39528094(1),4:39608882(1)	*UGDH-AS1*	1.45 × 10^−11^
RS191367039	chr4:148052195:G:A	Dense Area	4	4:148066298(1),4:148070999(1),4:148125334(1),4:148134899(1)	*ARHGAP10*	1.40 × 10^−10^
RS147570646	chr4:179504076:C:T	Dense Area	1	4:179560038(1)	*LOC105377563*/ *LOC124900818*	2.73 × 10^−9^
RS58470658	chr4:148160004:C:T	Dense Area	1	4:148166598(1)	*NR3C2*	4.57 × 10^−9^
RS116045382	chr4:11398948:C:T	Dense Area	0	NONE	*HS3ST1*	6.09 × 10^−9^
RS143507397	chr4:168554153:G:A	Dense Area	0	NONE	*PALLD*	1.89 × 10^−8^
RS186021694	chr4:173988596:A:G	Dense Area	3	4:173962614(1),4:173969459(1),4:173982430(1)	*LOC105377543*	3.22 × 10^−8^
RS113187843	chr5:138964719:T:A	Dense Area	0	NONE	*SIL1*	2.46 × 10^−9^
RS112818595	chr5:138863360:C:T	Dense Area	2	5:138813316(1),5:138850667(1)	*CTNNA1*	5.39 × 10^−9^
RS6912620	chr6:132552803:G:A	Dense Area	6	6:132460453(1),6:132479524(1),6:132480587(1),6:132551032(1),6:132551150(1),6:132592333(1)	*TAAR8*	1.08 × 10^−10^
RS59162058	chr7:5603217:C:T	Dense Area	2	7:5644533(1),7:5651116(1)	*FSCN1*	1.97 × 10^−8^
RS1657248	chr7:155112424:G:A	Dense Area	2	7:155112360(1),7:155113278(1)	*HTR5A*	3.64 × 10^−8^
RS114517045	chr8:77390103:T:G	Dense Area	5	8:77439758(1),8:77447273(1),8:77460647(1),8:77474685(1),8:77485921(1)	*LOC105375909*	2.20 × 10^−8^
RS189070945	chr10:66699780:C:T	Dense Area	11	10:66669847(1),10:66681835(1),10:66694622(1),10:66716457(1),10:66717173(1),10:66720546(1),10:66721909(1),10:66733490(1),10:66739808(1),10:66742213(1),10:66745040(1)	*CTNNA3*	1.76 × 10^−12^
RS59522962	chr10:66806298:T:C	Dense Area	1	10:66790114(1)	*CTNNA3*	1.58 × 10^−8^
RS79232684	chr11:47150163:G:A	Dense Area	0	NONE	*CSTPP1*	1.18 × 10^−8^
RS114694584	chr11:31973064:A:G	Dense Area	6	11:31930445(1),11:31935750(1),11:31939541(1),11:31948997(1),11:31955129(1),11:31986978(1)	*LOC110120941*/ *LOC107984420*	1.72 × 10^−8^
RS75439556	chr11:46775991:C:A	Dense Area	1	11:46802307(1)	*CKAP5*	2.95 × 10^−8^
RS184938993	chr12:56780836:G:A	Dense Area	1	12:56780646(1)	*HSD17B6*	4.02 × 10^−12^
RS150208861	chr14:97883545:C:T	Dense Area	0	NONE	*LINC01550*	1.61 × 10^−9^
RS150796751	chr17:38494521:G:A	Dense Area	0	NONE	*ARHGAP23*	3.82 × 10^−8^
RS143065709	chr18:57155409:C:T	Dense Area	2	18:57148687(1),18:57168501(1)	*BOD1L2*/*LINC02565*	4.17 × 10^−10^
RS73886707	chr22:46112492:C:T	Dense Area	3	22:46105565(1),22:46109178(1),22:46109365(1)	*MIRLET7BHG*	2.15 × 10^−9^
RS148811443	chr2:142145550:G:A	Percent Density	7	2:142055838(1),2:142158131(1),2:142161710(1),2:142167799(1),2:142215347(1),2:142218937(1),2:142242824(1)	*LRP1B*	1.56 × 10^−8^
RS79331071	chr4:166343624:C:T	Percent Density	26	4:166296011(1),4:166296276(1),4:166296612(1),4:166308772(1),4:166311509(1),4:166322829(1),4:166331896(1),4:166337499(1),4:166340457(1),4:166342254(1),4:166343104(1),4:166343888(1),4:166344382(1),4:166345601(1),4:166345825(1),4:166349179(1),4:166350417(1),4:166350715(1),4:166351544(1),4:166351573(1),4:166352076(1),4:166353507(1),4:166355475(1),4:166376134(1),4:166409358(1),4:166415758(1)	*LOC121056748*/ *LOC121056749*	4.41 × 10^−9^
RS138783664	chr9:114879160:C:T	Percent Density	1	9:114917870(1)	*LOC645266*/ *LOC124310630*	1.27 × 10^−8^
RS143877555	chr10:84298935:C:T	Percent Density	5	10:84294546(1),10:84344702(1),10:84348787(1),10:84352282(1),10:84366026(1)	*LINC00858*/*CCSER2*	1.01 × 10^−8^
RS145826214	chr13:99921286:C:T	Percent Density	3	13:99908944(1),13:99916943(1),13:99920000(1)	*CLYBL-AS3*	2.71 × 10^−8^

**Table 3 cancers-15-02776-t003:** Fine-mapping results with PIP ≥ 0.98. Epigenetic annotations: Repr = repressor, TssFlnkD = downstream flanking region to transcription start site, TssFlnkU = upstream flanking region to transcription start site, Het = heterochromatin.

Trait	Gene	Annotation	SNPID	PIP
Dense Area	*PDE10A*	Het	rs480268	0.98738
	*PDE10A*	Het	rs6907588	0.98244
	*PDE10A*	Het	rs576853	0.99956
	*PDE10A*	Het	rs481701	1
	*LOC101927404*		rs9967157	1
	*LOC101927404*		rs716961	1
	*LOC101927404*/*LOC105372168*		rs34217531	1
	*LOC101927404*/*LOC105372168*		rs1306871	1
	*LOC105372168*		rs9954012	1
	*LOC105372168*	ncRNA_gene	rs611750	1
	*LOC105372310*	Het	rs10412042	1
	*LOC105372310*	Het	rs12462802	1
	*LOC105372310*	Het	rs7253843	1
	*LOC105372310*	Het	rs4277458	1
	*LOC100129265*/*BNIP3P19*	Het	rs28493283	1
	*BNIP3P19*/*BNIP3P20*	Het	rs10426611	1
	*BNIP3P20*/*BNIP3P21*	pseudogene	rs9989730	1
	*BNIP3P20/BNIP3P21*	pseudogene	rs28786195	1
Percent Density	*SH3GL3*	ncRNA_gene	rs10906974	1
	*SH3GL3*	ncRNA_gene	rs301847	1
	*SH3GL3*	ncRNA_gene	rs6602974	1
	*SH3GL3*	lnc_RNA	rs55641568	1
	*SH3GL3*	lnc_RNA	rs11853676	1
	*SH3GL3*	lnc_RNA	rs7350762	1
	*SH3GL3*	lnc_RNA	rs12905964	1
	*KIFC3*	mRNA	rs1582594	1
	*KIFC3*	mRNA	rs2967139	1
	*KIFC3*	EnhA1	rs4784864	1
	*KIFC3*	EnhA1	rs2911348	1
	*KIFC3*	EnhA2	rs2967137	1
	*KIFC3*	EnhA2	rs59350294	1
	*KIFC3*	mRNA	rs140234666	1
	*KIFC3*	Het	rs9938048	0.99999
	*KIFC3*/*CNGB1*	EnhA1	rs838583	1
	*CNGB1*	mRNA	rs691656	1

**Table 4 cancers-15-02776-t004:** Comparison of results with effect estimates and significance reported in a recently published breast density GWAS for 27,900 European ancestry individuals. Summary statistics comparison with SNPs found to be significant from Chen et al. manuscript. MAFs refer to minor allele frequency from this study.

	Breast Area			Percent Density			Dense Area		
	A1_FREQ	BETA	*p*	A1_FREQ	BETA	*p*	A1_FREQ	BETA	*p*
rs11205303	0.078	−0.047	0.447	0.077	−0.053	0.481	0.077	−0.083	0.310
rs1868992	0.412	0.058	0.160	0.413	0.030	0.547	0.412	0.048	0.370
rs17625845	0.069	0.154	0.019	0.068	0.055	0.486	0.068	0.103	0.229
rs6851733	0.093	−0.038	0.496	0.095	0.018	0.786	0.094	−0.016	0.824
rs413472	0.266	0.057	0.123	0.266	−0.045	0.311	0.266	−0.028	0.557
rs335189	0.167	−0.004	0.927	0.167	0.032	0.533	0.168	0.093	0.093
rs11745230	0.402	0.021	0.521	0.401	−0.016	0.696	0.402	0.008	0.859
rs2112670	0.171	−0.035	0.410	0.171	−0.026	0.612	0.171	−0.054	0.327
rs2042239	0.154	−0.013	0.766	0.152	−0.074	0.168	0.154	−0.061	0.289
rs3819405	0.460	−0.003	0.936	0.456	−0.125	0.002	0.456	−0.106	0.015
rs4897107	0.159	−0.049	0.257	0.160	−0.055	0.287	0.160	−0.153	0.007
rs9397436	0.069	0.043	0.490	0.068	−0.048	0.524	0.068	0.077	0.344
**rs16885613**	**0.356**	**−0.115**	**0.001**	**0.356**	**0.197**	**8.07 × 10^−7^**	**0.357**	**0.117**	**0.007**
**rs10087804**	**0.296**	**−0.113**	**0.001**	**0.296**	**0.209**	**9.17 × 10^−7^**	**0.297**	**0.146**	**0.002**
rs58847541	0.290	−0.009	0.792	0.288	−0.036	0.384	0.290	−0.017	0.700
rs2138555	0.315	−0.032	0.355	0.318	−0.034	0.416	0.318	−0.047	0.295
rs10995187	0.071	−0.018	0.773	0.070	−0.012	0.876	0.070	0.014	0.864
rs4980383	0.274	−0.002	0.956	0.275	0.055	0.194	0.275	0.020	0.671
rs11836164	0.221	0.026	0.510	0.221	−0.003	0.953	0.221	0.011	0.835
rs7297051	0.141	−0.004	0.929	0.141	0.010	0.852	0.142	0.014	0.802
rs61938093	0.209	0.132	0.001	0.211	−0.107	0.022	0.210	−0.038	0.451
rs4499190	0.150	−0.070	0.137	0.151	−0.050	0.377	0.151	−0.089	0.145
rs11646715	0.166	−0.051	0.234	0.168	0.029	0.573	0.168	−0.005	0.931
rs12462111	0.169	0.063	0.139	0.169	−0.022	0.658	0.169	0.033	0.549
rs1231281	0.263	0.038	0.306	0.263	0.025	0.580	0.263	0.040	0.412
rs17789629	0.040	−0.039	0.639	0.041	0.132	0.185	0.040	−0.022	0.836
rs34066050	0.209	−0.018	0.661	0.208	0.012	0.799	0.209	0.037	0.476
rs73169097	0.056	−0.104	0.132	0.057	−0.016	0.843	0.057	−0.045	0.615

Bolded rows are statitsically signficant at *p* < 0.001.

## Data Availability

The data presented in this study are available on request from the corresponding author. The data are not publicly available due to privacy considerations.

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
