# Peer review of "Genome-Wide Association Study of Breast Density among Women of African Ancestry"

_cancers, 2023, doi:10.3390/cancers15102776_

Round 1
Reviewer 1 Report
The authors conducted a GWAS of mammographic breast density in Black women using data from the Penn Medicine BioBank. It is a strong study and a well-written manuscript. I offer a few questions/suggestions that could further enhance the paper.
1. Given that Black or African American individuals are a genetically admixed population, how was that accounted for or evaluated in this analysis? Can you provide more information on the principal components analysis to determine population structure? Was admixture mapping considered or are there plans for subsequent analyses?
2. Why did you require all four views? Some studies require only CC or CC and MLO from one breast.
3. This is a small population for a GWAS study. I’m surprised so many variants reached genome wide significance. This suggests that they are very strongly associated with MD. Authors should discuss their limited ability to identify more modest associations and the implications for their results.
4. Minor: the order of exclusions seems odd to me. Why not start with a population of Black or African American and then apply the other data exclusions?
Author Response
- Given that Black or African American individuals are a genetically admixed population, how was that accounted for or evaluated in this analysis? Can you provide more information on the principal components analysis to determine population structure? Was admixture mapping considered or are there plans for subsequent analyses?
Thank you for your comment. We understand your concern about the potential impact of genetic admixture on our results. To account for population structure in our analysis, we performed principal component analysis (PCA) on our study population and projected it onto the 1000 Genomes reference dataset. We used all significant principal components in our genome-wide association study (GWAS). We have now included the PCA plot and scree plot as Supplemental Figure 1 to provide more details about our population structure analysis. While we did not run admixture mapping in this study, we do plan to conduct further analyses to explore the potential effects of admixture on our results.
- Why did you require all four views? Some studies require only CC or CC and MLO from one breast.
The LIBRA software quantifies dense area and breast area across all available views. Since each view is only a 2D cross section of the breast, none of the views independently capture the true volume of density in the 3D breast. By averaging our estimates of density across each view for each woman, we produce a more robust estimate of the actual density in the breast. We have added this justification in the methods section 2.2.
- This is a small population for a GWAS study. I’m surprised so many variants reached genome wide significance. This suggests that they are very strongly associated with MD. Authors should discuss their limited ability to identify more modest associations and the implications for their results.
We acknowledge that our study population is relatively small for a GWAS, and we appreciate your concern about the potential impact on our ability to identify more modest associations. It is important to note that while our sample size is small, breast density was measured quantitatively using an automated algorithm with high accuracy. In addition, breast density is known to be highly heritable. In combination, the continuous quantitative trait and the high heritability may have resulted in the ability to detect associations despite a relatively small sample size. However, we agree that the small sample size limits our ability to detect more modest associations. We have added these points to the discussion section and acknowledged the need for replication in larger populations and the importance of conducting meta-analyses to increase statistical power.
- Minor: the order of exclusions seems odd to me. Why not start with a population of Black or African American and then apply the other data exclusions?
We utilized a population that was initially assembled for another study, for which the exclusions were applied in the order described.
Reviewer 2 Report
This paper highlight the importance of breast density GWAS among diverse populations, and also provide a novel insights into genetic factors associated with breast density and help elucidate mechanisms by which density increases breast cancer risk. But the following questions should be clearly described in the article. 1. Since the incidence of breast cancer tends to be more younger, it would be more valuable to select the younger population for analysis. But the paper chose a mean age of 56.45 for analysis, which is too old. 2. This paper carries out a comparative analysis of breast density in Black women and White women. In the satistical data, whether the data of White women can be supplemented for comparison, so that the results will be more striking. 3. When performing the polygene association analysis for breast density, it be better to supplemented the links that have been reported for breast cancer-related genes or signaling pathways.
Author Response
- Since the incidence of breast cancer tends to be more younger, it would be more valuable to select the younger population for analysis. But the paper chose a mean age of 56.45 for analysis, which is too old.
We agree that generalizability of our findings are important. The median age of breast cancer diagnosis in the U.S. is 63 years (https://seer.cancer.gov/statfacts/html/breast.html). Breast density does tend to be greater for younger rather than older individuals, however, even among older women breast density is predictive of breast cancer risk. (https://jamanetwork.com/journals/jamanetworkopen/fullarticle/2783508). We therefore chose to include a broad age group in our study, reflective of the population of women who typically undergo mammography screening. We believe that our results will therefore generalize to women across age groups.
- This paper carries out a comparative analysis of breast density in Black women and White women. In the satistical data, whether the data of White women can be supplemented for comparison, so that the results will be more striking.
Given that there have been several larger previous GWAS studies of breast density among women of European ancestry previously published, we did not think that including White women in this analysis would be useful. The purpose of the study was to perform the first breast density GWAS among African ancestry women to help rectify the underrepresentation of this group in prior GWAS studies.
- When performing the polygene association analysis for breast density, it be better to supplemented the links that have been reported for breast cancer-related genes or signaling pathways.
Thank you for your comment. We agree with your suggestion that supplementing the polygene association analysis with reported links to breast cancer-related genes or signaling pathways may provide additional insights into the genetic basis of breast density. In this study, we investigated the overlap between our top-associated variants and those reported in previous studies of breast cancer and other related traits. We believe that this integrative approach can help to shed light on the underlying biology of breast density and its relationship to breast cancer risk. While we did not specifically explore the reported links between breast cancer-related genes or signaling pathways and breast density in our study, we agree that this would be a worthwhile avenue for future research. We have added to the discussion that further research is needed to validate these findings and to explore the functional implications of the identified genetic variants.